# Elafin as a Prognostic Marker in Esophageal Squamous Cell Carcinoma: A Pilot Study Using Three-Dimensional Imaging and Genomic Profiling

**DOI:** 10.3390/cancers15153825

**Published:** 2023-07-27

**Authors:** Wei-Chung Chen, Chun-Chieh Wu, Yu-Peng Liu, Guan-Yu Zhuo, Yao-Kuang Wang, Yi-Hsun Chen, Chu-Chih Chen, Yin-Han Wang, Ming-Tsang Wu, I-Chen Wu

**Affiliations:** 1Ph.D. Program in Environmental and Occupational Medicine, Kaohsiung Medical University, Kaohsiung 807, Taiwan; u103803001@kmu.edu.tw (W.-C.C.); 960021@cc.kmuh.org.tw (M.-T.W.); 2Department of Medicine, Faculty of Medicine, College of Medicine, Kaohsiung Medical University, Kaohsiung 807, Taiwan; wucchieh@kmu.edu.tw (C.-C.W.); 970395@kmuh.org.tw (Y.-K.W.); 3Department of Pathology, Kaohsiung Medical University Hospital, Kaohsiung Medical University, Kaohsiung 807, Taiwan; 4Graduate Institute of Clinical Medicine, Kaohsiung Medical University, Kaohsiung 807, Taiwan; ypliu@kmu.edu.tw; 5Institute of New Drug Development, China Medical University, Taichung 404, Taiwan; zhuo0929@mail.cmu.edu.tw; 6Division of Gastroenterology, Department of Internal Medicine, Kaohsiung Medical University Hospital, Kaohsiung 807, Taiwan; 1020420@kmuh.org.tw; 7Division of Biostatistics and Bioinformatics, Institute of Population Health Sciences, National Health Research Institutes, Miaoli 350, Taiwan; ccchen@nhri.edu.tw (C.-C.C.); yhwang27@nhri.edu.tw (Y.-H.W.); 8Department of Family Medicine, Kaohsiung Medical University Hospital, Kaohsiung Medical University, Kaohsiung 807, Taiwan; 9Center for Cancer Research, Kaohsiung Medical University, Kaohsiung 807, Taiwan

**Keywords:** esophageal squamous cell carcinoma, elafin, three-dimensional imaging, prognosis

## Abstract

**Simple Summary:**

This research leveraged three-dimensional confocal imaging to examine elafin’s spatial distribution in locoregional esophageal squamous cell carcinoma (ESCC) tumors. Our findings highlighted distinct elafin localizations in patients with poor versus favorable prognoses. Elafin expression was significantly higher in tumor regions, contributing to a poorer prognosis. Complementary in vitro studies revealed that elafin promotes ESCC cell proliferation, migration, and invasion via the epithelial–mesenchymal transition pathway. These findings suggest that the targeted inhibition of elafin could potentially serve as a novel therapeutic approach aimed at improving survival rates in patients with locoregional ESCC.

**Abstract:**

Esophageal cancers are globally the sixth deadliest malignancy, with limited curative options. The association of high serum elafin levels, a molecule produced by epithelial cells, with esophageal squamous cell carcinoma (ESCC) risk is established, but its link to poor ESCC prognosis remains unclear. To explore this question, we first used three-dimensional confocal imaging to create a model of the spatial distribution of elafin inside locoregional ESCC tissues. Then, after analyzing data obtained from whole-genome microarrays for ESCC cell lines and their more invasive sublines, we performed in vitro experiments using RNA sequencing to identify possible elafin-related pathways. Three-dimensional tissue imaging showed elafin distributed as an interweaved-like fibrous structure in the stroma of tissue obtained from patients with high serum levels of elafin and poorer prognoses. By contrast, the signal was confined inside or around the tumor nest in patients who had lower serum levels and better survival. The analysis of a TCGA dataset revealed that higher levels of elafin mRNA in stage I–IIIA ESCC patients were associated with shorter survival. The in vitro studies revealed that elafin promoted ESCC cell proliferation, migration, and invasion via the epithelial–mesenchymal transition pathway. Thus, elafin inhibition could potentially be used therapeutically to improve survival in patients with locoregional ESCC.

## 1. Introduction

Esophageal squamous cell carcinoma (ESCC) is the most common histological type of esophageal malignancy in Asian countries and worldwide [1,2]. This malignancy is a highly aggressive tumor, and once diagnosed, most cases are already locally advanced or metastasized and prognosis is poor [3]. Besides several serum tumor markers such as carcinoembryonic antigen (CEA), squamous cell cancer antigen (SCCA), and tissue polypeptide antigen (TPA) being used as serum tumor markers clinically, we recently reported that serum elafin can serve as another potential prognostic marker in ESCC patients with stage I–IIIA tumors [4,5,6].

Elafin, also known as peptidase inhibitor 3 (PI3) or skin-derived anti-leukoproteinase (SKALP), is a secretory small molecule [7,8]. It is produced by epithelial cells and is mainly involved in anti-microbial and anti-inflammatory functions [9,10,11]. Although several studies have found the expression of elafin to be higher in squamous cell carcinoma tissues of the lung, head and neck, and esophagus than their adjacent normal tissue parts [7,8,12], little is known about the mechanisms underlying its effect on prognosis.

Most conventional pathological analyses and clinical cancer studies have been conducted using two-dimensional (2-D) formalin-fixed paraffin-embedded (FFPE) tissue sections. The information it provides is limited to the tissue sections with a thickness of half a cell (4–5 μm) and does not adequately provide a comprehensive representation of tumor heterogeneity. Three-dimensional (3-D) imaging can provide a more panoramic view of tissue structure than the 2-D planar view and a better picture of tumor development, differentiation, and invasion [13,14]. Therefore, we performed 3-D constructive tissue imaging to observe the histological morphology and the distribution of secretory elafin protein in tumor tissues collected from locoregional ESCC patients and in vitro studies to investigate the mechanisms underlying the association between the distribution of elafin and prognosis.

## 2. Materials and Methods

### 2.1. Selection of ESCC Patients to Provide Sample for 3-D Imaging

For the 3-D imaging study, we enrolled two paired-case patients who had received esophagectomies and who had enough tissue for samples at least 4 mm thick in archived tissue blocks from one previous study cohort of 119 incident ESCC patients [5]. In brief, the previous study cohort consisted of potentially eligible incident and pathologically proved ESCC patients with stage I–IIIA tumors, recruited between 2000 and 2016, and from Kaohsiung Medical University Hospital (KMUH) and Kaohsiung Veterans General Hospital (KVGH), two medical centers in southern Taiwan. A detailed flowchart of our selection process is presented in Appendix A.

### 2.2. Sample Preparation for 3-D Imaging and Confocal Imaging Acquisition

The detail pipeline of this step is illustrated in Appendix A. One tissue sample of 4 μm thickness was first sectioned from one archived tissue block for hematoxylin and eosin (HE) staining to locate the tumorous part. A cylindrical-shaped tissue sample 4 mm in diameter and 4 mm thick was harvested from the tissue block with the use of punching biopsy needles. The 4 mm thick side of the cylindrical-shaped tissue was cut into a series of small cylindrical slices, each measuring 150 μm thick. Subsequently, the small cylinder tissues (4 mm in diameter and 150 μm thick) were deparaffinized using Hemo-De reagent (Cat. HD150; HEMO—DE, Inc.; Keller, TX, USA) at 37 °C for 2 h and then maintained at room temperature for 1 hr. The samples were then washed using a series of solutions containing decreasing concentrations of alcohol. Finally, the samples were washed in PBS overnight to finish the deparaffinization process.

The deparaffinized samples were processed with antigen retrieval and blocking reagents and then stained with SYTO 16 (Cat. S7578, Thermo Fisher Scientific, Waltham, MA, USA) to label the cell nucleus and stained with DiD (Cat. V22887, Thermo Fisher Scientific, Waltham, MA, USA) to label the cell membrane and cytoplasm. After staining, the deparaffinized samples were relabeled elafin, which was achieved by treating each sample with mouse anti-elafin primary antibody (1:40, catalog number HM2063, HycultBiotech, Wayne, PA, USA) at 4 °C for ~72 h followed by treatment with Alexa Fluor™ goat anti-mouse IgG secondary antibody (Cat. B40913; Thermo Fisher Scientific, Waltham, MA, USA) at 4 °C for one day, followed by treatment with Alexa Fluor™ tyramide reagent (Cat. B40913; Thermo Fisher Scientific, Waltham, MA, USA) at 25 °C for 30 min.

The labeled specimens were then immersed in optical clear solution (Focus Clear, Cat. FC-101; CelExplorer, Hsinchu, Taiwan), which has a refractive index of about 1.45 at room temperature for 15 h to obtain a clear and labeled specimen [15,16]. Three-dimensional histopathology imaging was performed under Olympus FV3000 laser scanning confocal microscopy with a 40× air objective lens (UPlanXApo, Olympus Microscopy, Tokyo, Japan). Sample scanning was set at 512 × 512 pixels (equaling 317.95 µm × 317.95 µm) at the x/y plane in 0.7 µm increments at the z axis [13,17]. The images were acquired and detected under wavelength excitation and the emissions set as follows: 488 nm and 530 nm for SYTO 16; 561 nm and 605 nm for elafin; and 640 nm and 692 nm for DiD.

### 2.3. Processing of 3-D Tissue Images and 3-D Tissue Modeling

The detail pipeline of this step is illustrated in Appendix A. The images captured by the 40× air objective lens were stitched together to create a gross tissue view by Imaris Stitcher version 9.7.0 (Oxford Instruments, Zurich, Switzerland). The stitched file was then uploaded into FIJI software version 1.53c for image processing [18]. The signals of elafin, DiD (for cell membrane), and SYTO 16 (for cell nucleus) in each layer were subtracted from the background intensity to reduce noise and were further filtered with median signal intensity to obtain a sharper boundary for each signal. For the pseudo-color images, green was assigned to elafin, blue to cell membrane and cytoplasm, and red to the cell nucleus. After the colors were assigned, the pixels of each color in the same layer were counted to assess the relative quantities of elafin vs. DiD or SYTO 16. Every tissue layer of each sample was compiled into a 3-D structure, which was then collapsed into a single 2-D layer using the Z projection tool. This method resulted in an image reflecting the sum of the intensities from all layers, providing an overall depiction of signal intensity. A representative gross view of pair 1 had a 100% resolution scale, but that of pair 2 was limited to a 50% resolution scale due to hardware limitations.

For the 3-D modeling, we used Imaris software version 9.5 (Oxford Instruments, Zurich, Switzerland) to build, render, and calculate the elafin model. After processing with background subtraction and a median filter, the elafin and cell nuclear models were created by surface and spot rendering, respectively. In each tissue layer, the relative elafin intensity was calculated as the pixel numbers of the elafin signal divided by the pixel numbers of the cell membrane signal. The cutoff value of the relative elafin intensity was set as 0.01 to filter out weak signals and select tissue layers with relative elafin intensity > 0.01 for further 3-D surface rendering (Table 1).

### 2.4. Elafin Expression in Tissue Arrays of Resected ESCC Specimens

The tissue arrays were made of FFPE surgical specimens of the tumor and distant normal esophageal tissues of another 110 ESCC patients collected from the archived tissue banks of KMUH and China Medical University Hospital during 2005–2012. We selected the archived tissue samples of 63 ESCC patients with stage I–IIIA for the subsequent immunohistochemistry (IHC) staining. Because personal information was encrypted, written informed consent was waived.

In the slide tissue array, one slide contained two cancer tissues and two normal tissues, with each paring collected from the same subject (15 patients in total). The specimens were deparaffinized and the tissue blocks were autoclaved with the target retrieval solution (Cat. S2368; Target Retrieval Solution, pH 9; Agilent Technologies, Santa Clara, CA, USA). Elafin antibody (1:150; catalog number HM2063, HycultBiotech, Wayne, PA, USA) was mounted and incubated on the specimens. Then, anti-mouse/rabbit secondary antibody conjugated with HRP (Cat. K5007; ChemMate™ DAKO EnVision™ Detection Kit, Agilent Technologies, Santa Clara, CA, USA) was added. All slides were lightly counterstained with Mayer’s hematoxylin (Cat. 30005; MUTO PURE CHEMICALS, Tokyo, Japan). The elafin staining of the specimens obtained from each patient was accessed and scored under high-power field (200×) by a pathologist (Dr. CC Wu) who was blinded to the patients’ clinical status. The intensity of elafin staining in the cancer cells and normal tissues was scored separately, as follows: 0–1 = negative or focal weak cytoplasmic staining; 2 = diffuse moderate cytoplasmic staining and blurred membrane staining; and 3 = diffuse strong cytoplasmic staining and distinct membrane staining (Appendix A).

### 2.5. Elafin mRNA Expression in TCGA Dataset

The mRNA expression and clinical data were acquired from The Cancer Genome Atlas (TCGA) Esophageal Cancer project by the UCSC Xena platform (http://xena.ucsc.edu/, accessed on 11 July 2019) shared from The Cancer Genome Atlas. From the UCSC Xena platform, we chose the database of GDC TCGA Esophageal Cancer (GDC TCGA ESCA), version “09-14-2017”. The type of gene expression data was RNA sequencing with log2 (fpkm-uq+1) as the expression unit. There were 185 esophageal cancer patients enrolled in the GDC TCGA-ESCA project originally, and 161 remained after clearing the blank identifiers of gene expression, gender, height, weight, age, tumor stage, survival and follow-up time, and smoking and alcohol consumption history. Finally, we analyzed the relationship of elafin (PI3; HGNC:8947) mRNA levels with the survival in 93 stage I–IIIA esophageal cancer patients (59 ESCC and 34 esophageal adenocarcinoma) who received esophagectomy without prior cancer treatment from the TCGA dataset (version 09-14-2017).

### 2.6. Cell Lines and Cell Culture

The details regarding the cell lines used and how they were cultured has been previously described [19]. Briefly, CE81T, KYSE270, and OE21 cell lines were obtained from the Food Industry Research and Development Institute (Hsinchu, Taiwan). CE81T and OE21 cell lines and their more aggressive sublines established previously were also used [19]. The cells were maintained in Dulbecco’s Modified Eagle Medium (DMEM) supplemented with 10% fetal bovine serum (FBS), 1% non-essential amino acids (NEAA), 100 U/mL penicillin, and 100 μg/mL streptomycin. The KYSE270 cells were cultured in DMEM/F-12 medium supplemented with 2% FBS, 100 U/mL penicillin, and 100 μg/mL streptomycin, while the OE21 cells were cultured in RPMI-1640 medium supplemented with 10% FBS, 100 U/mL penicillin, and 100 μg/mL streptomycin. All of the products used for our cell cultures were purchased from Thermo Fisher Scientific, Inc. (Waltham, MA, USA). The cell lines were kept in a humidified incubator under 5% CO_2_ at 37 °C.

### 2.7. Plasmids, Short Hairpin RNAs, and Lentivirus Production

To generate the elafin-expressing construct, we synthesized the customized DNA constructs of human elafin (GenBank: L10343.1) with the use of GeneArt (Thermo Fisher Scientific, Waltham, MA, USA). The myc-tagged elafin was generated by cloning the cDNA with the C-terminal myc sequence into the pLEX-MCS lentiviral vector (Thermo Fisher Scientific, Waltham, MA, USA). The two clones of shRNAs targeting human elafin: shRNA#1 (TRCN0000073667/NM_002638), shRNA#6 (TRCN0000373198/NM_002638), and a non-specific scramble shRNA sequence, were purchased from the National RNAi Core Facility at Academia Sinica, Taipei, Taiwan. The lentivirus particles were prepared by co-transfecting gene-expressing or shRNA lentiviral plasmids with psPAX2 and pMD2.G plasmids into HEK293T cells (psPAX2: Addgene plasmid # 12260; pMD2.G: Addgene plasmid # 12259, Watertown, MA, USA). The infection of cells was carried out in the presence of 10 μg/mL polybrene for 48 h and then puromycin (1 μg/mL) was added to the cells for another 48 h to select cells stably expressing elafin and cells in which elafin had been knocked down.

### 2.8. Quantitative Real-Time RT-PCR (TaqMan Assay)

Total RNA was extracted using RNeasy Micro Kit (Cat. 74404; QIAGEN, Venlo, The Netherlands) and cDNA was synthesized by GoScript™ Reverse Transcriptase (Cat. A2791, Promega, Madison, WI, USA). Elafin expression was quantified using Taqman gene expression assay reagents (Assay ID: Hs00160066; Thermo Fisher Scientific, Waltham, MA, USA) on an ABI Prism 7900 System following the manufacturer’s directions.

### 2.9. Immunoblot Analysis

All cell lysates were quantified and resolved on a sodium dodecyl sulfate–polyacrylamide gel electrophoresis gel, which was transferred onto a polyvinylidene fluoride membrane (Cat. 88518; Millipore, Burlington, MA, USA). The protein markers we used were PageRuler™ Prestained Protein Ladder (Cat. 26616; Thermo Fisher Scientific, Waltham, MA, USA) and Novex Sharp Pre-Stained Protein Standard (Cat. LC5800; Thermo Fisher Scientific, Waltham, MA, USA). The membranes were incubated with the indicated primary antibodies (listed below), followed by horseradish peroxidase-conjugated secondary antibodies (1:20000; anti-mouse: Cat. 115-035-003; anti-rabbit: Cat. 111-035-003; Jackson Immu-noResearch, West Grove, PA, USA) and an enhanced chemiluminescence solution (NEN Life Science, Boston, MA, USA). The following primary antibodies were used: Elafin (1:500; Cat. HM2063; Hycult Biotech, Wayne, PA, USA), c-Myc (1:1000; Cat. sc-40; Santa Cruz Biotechnology, Dallas, TX, USA), Snail (1:1000; Cat. GTX125918; GeneTex, Irvine, CA, USA), Slug (1:1000; Cat. GTX128796; GeneTex, Irvine, CA, USA), E-Cadherin (1:1000; Cat. GTX100443; GeneTex, Irvine, CA, USA), N-Cadherin (1:1000; Cat. ab76011; Abcam, Cambridge, MA, USA), and β-actin (1:1000; Cat. 4970; Cell Signaling Technology, Danvers, MA, USA).

### 2.10. Cell Proliferation Assay

Stable clones of pLEX-vector and pLEX-elafin cells were plated on six-well plates (2 × 10^5^ cells/well) and stained with trypan blue. The proliferation of stained cells was assessed by counting the total viable cell numbers at different times.

### 2.11. Cell Motility and Invasion Assays

Migration and invasion assays were conducted in 24-well Corning Hanging Inserts (Cat. 3422, Corning Incorporated, Corning, NY, USA) and 24-well BioCoat Matrigel Invasion Chambers (Cat. 354480; Corning Incorporated, Corning, NY, USA), respectively. Cells resuspended in 300 μL serum-free medium were added to the top chamber (1 × 10^5^ cells/well) and medium supplemented with DMEM/10% FBS was added to the bottom chamber as a chemoattractant. Following 16–18 h incubation at 37 °C, cells that migrated or invaded through the membrane (migration) or Matrigel (invasion) were fixed and stained with 0.1% crystal violet (Cat. C0775; Sigma-Aldrich, St. Louis, MO, USA). The number of cells was counted in three random fields under a 100× objective lens.

Cell migration was assessed by wound healing using IBIDI Culture-Inserts (Cat. 80241; Ibidi GmbH, Gräfelfing, Germany). Briefly, cells (3 × 10^5^ cells/70μL/well) were seeded into each well of a 24-well tray with culture inserts. The culture inserts were then removed, the cell debris was removed by washing with PBS, and the cells were cultured in DMEM supplemented with 10% FBS, 1% NEAA, 100 U/mL penicillin, and 100 μg/mL streptomycin. Images were captured and analyzed after wounding based on the distance the cell monolayer had migrated as the wound healed at 0, 16, and 48 h. The wound closure rate was equal to the recovered distance divided by the original width of the scratch.

### 2.12. Statistical Analysis

To compare the relative elafin intensity of two pairs (E365 vs. E385 and E535 vs. E421), we performed the permutation test for each pair separately. First, the tissue layers obtained from each patient were grouped into 20 clusters with approximately 10 layers in each cluster. A cutoff of 0.05 was set to dichotomize the elafin intensities into 0 (<0.05) and 1 (≥0.05). Under the null hypothesis of no difference, the observed mean difference of the elafin intensity scores of the clusters would be insignificant in a total of 220 (=1,048,576) reallocations of the paired observations of the clusters. The *p*-value was obtained by the observed ranking among all possible mean differences of the reallocations from the smallest to the largest. The midpoint of the tied ranks was adopted in deriving the *p*-value.

For the elafin expression of IHC staining in the slide tissue array of resected ESCC specimens, a Student’s t-test or Wilcoxon rank-sum test was performed to compare the difference of elafin expressions in tumor vs. normal tissues. For elafin mRNA expression in the TCGA dataset, we first determined the optimal cutoff elafin mRNA level. We used a Cox regression model to fit for the patients’ survival periods classified by the elafin cutoff levels that were sorted sequentially from the lowest to the highest. Other covariables, including age, sex, smoking status, and alcohol consumption, were also adjusted for the model. The elafin cutoff level that yielded the maximum Youden’s index (=sensitivity + specificity − 1) from the resultant Cox regression model fitting was chosen to be the optimal cutoff of elafin mRNA levels.

For the in vitro experiments, a Student’s *t*-test was used to compare the difference between the experimental groups. All *p*-values were two-sided, and significance was defined as *p*-value < 0.05.

## 3. Results

### 3.1. Selection of Locoregional ESCC Patients for 3-D Tissue Imaging

Of the 119 ESCC patients we included in this study, 17 had received esophagectomies with enough tumor tissues to create blocks for 3-D imaging (Appendix A). Using the mean elafin levels and mean survival times, we subcategorized these 17 eligible patients into four groups and then selected four patients to create two pairs matched by gender (male), clinical staging, and treatment modality (Figure 1A). Subjects E365 and E535 had high serum elafin levels, but poor prognosis, whereas Subjects E385 and E421 had low serum elafin levels, but favorable prognosis (Table 1).

### 3.2. Distinct 3-D Spatial Distribution of Elafin Expression in ESCC Patients with Different Prognoses

Using laser scanning confocal microscopy, we obtained ~200 signal intensity values (one value for each 0.7 μm thick tissue layer), ranging from 182 to 215 tissue layers, from each of the two pairs (Figure 1B–D; Table 1). The spatial distribution patterns of relative elafin intensity in E365 and E535 distinctly differed from those of E385 and E421, with the single peak of relative elafin intensity present in E365 and E535, but not in E385 and E421. The distributions of elafin signals in different 2-D tissue layers were widely heterogeneous within the same subjects and across the four subjects (Figure 1E,F).

Using the permutation tests, the observed mean differences in the clustered elafin intensity scores for pair 1 (E365 vs. E385) and pair 2 (E535 vs. E421) were 0.160 and 0.195, with the smallest rankings of 983,041 and 1,015,809 out of all possible re-allocations (1,048,576), respectively. Therefore, both *p*-values of the differences of the two pairs were significant, which were 0.03 (=1 − (983,041 + 65,536/2)/1,048,576) and 0.016 (=1 − (1,015,809 + 32,768/2)/1,048,576), respectively. Thus, the overall relative elafin intensities were significantly higher in E365 and E535, when compared to those in E385 and E421 (Appendix A).

When we projected the optical images of all ~200 tissue layers (the 3-D structure) into one single 2-D layer structure, the elafin expressions were diffusely and randomly distributed throughout the entire stroma region in E365 and E535 (Figure 2A,G). By contrast, in E385 and E421, the elafin expressions showed sphere- and stripe-like shapes and were localized within the tumor nest region (Figure 2C,J). Imaris software 3-D rendering of the elafin signal only showed that elafin forms a complex web extending across the stroma in E365 and E535 (Figure 2B,H; Appendix A), but not in E385 and E421. Interestingly, elafin was concentrated within the tumor nest and did not infiltrate into the stroma in E385 and E421 (Figure 2E,K; Appendix A).

### 3.3. Overexpression of Elafin Protein in Tissue Arrays of Resected ESCC Specimens

To examine tumor tissue elafin expression in 63 ESCC patients with stage I–IIIA, we found that the elafin expression levels were higher in the tumor tissues than in their adjacent normal tissues (1.72 ± 0.97 vs. 1.17 ± 0.52; *p* < 0.0001, Appendix A). Excluded elafin expressions were missing in five tumor parts and 22 normal parts, and the difference remained significant in 36 patients from whom we were able to collect tumor-normal paired specimens (difference = 0.53 ± 1.10, *p* = 0.006; Figure 3A).

Additionally, using the clinical data of 93 eligible esophageal cancer patients with stage I–IIIA from the TCGA dataset (version 09-14-2017), we found that the optimal cutoff of the elafin mRNA level was 25 (maximum Youden’s index = 0.086 and the corresponding *p*-value for group difference was 0.03). Dichotomized by 25 of the elafin mRNA levels, patients with high elafin mRNA levels had a shorter survival period than those with low elafin expression (*p*-value = 0.015; Figure 3B).

### 3.4. Elafin Increased Proliferation, Migration, and Invasion of the ESCC Cell Lines

Myc-tagged elafin was ectopically and stably expressed in the CE81T2 and KYSE270 cell lines, which were two cell lines found in a previous study to have relatively lower expressions of elafin RNA and protein levels [5]. The forced expression of elafin-myc increased the proliferation of CE81T2 and KYSE270 cells compared with the myc-control cells (Figure 4A; Appendix A). The in vitro migration assay showed that the overexpression of elafin-myc significantly increased the trophic factor-triggered migration of CE81T2 and KYSE270 cells (Figure 4B). In addition, the overexpression of elafin-myc increased the invasion ability of CE81T2 and KYSE270 cells (Figure 4C). the overexpression of elafin-myc also increased the directional migration of CE81T2 and KYSE270 cells in the wound-healing assays (Figure 4D).

### 3.5. Elafin Induced Epithelial-to-Mesenchymal Transition (EMT)

To study the possible mechanism underlying the elafin-promoted migration and invasion of ESCC cells, we performed RNA sequencing of NGS to examine the gene expression profiles of the elafin-overexpressed CE81T2 cell line and the elafin-knockdown CE81T2-4 cell lines (please see Appendix A). Using a cutoff of a log2 fold change equaling a ratio >1 or <−1, we found 922 genes to be concurrently upregulated in the elafin-overexpressed CE81T2 cell line and downregulated in the elafin-knockdown CE81T2-4 cell line (Figure 5A). The ingenuity pathway analysis (IPA) of the molecular and cellular functions of these 922 genes identified the functional pathways underlying molecular transport, cellular movement, cell signaling, and molecular metabolism (Figure 5B), further supporting the role of elafin in ESCC invasion and metastasis. Furthermore, the mesenchymal markers were upregulated, and the epithelial markers were downregulated after overexpressing the elafin CE81T2 cell line and vice versa after knocking down elafin in the CE81T2-4 cell line, suggesting that elafin induced EMT (Figure 5C; Appendix A). The overexpression of elafin by elafin-myc in both the CE81T2 and KYSE270 cell lines decreased the protein levels of E-cadherin and increased the protein levels of N-cadherin, Snail, and Slug in both the CE81T2 and KYSE270 cell lines (Figure 5D; Appendix A). Additionally, knocking down elafin by elafin-specific shRNAs showed the opposite results in the CE81T2-4 and OE21 cell lines (Appendix A). By using the most efficient clone shRNA#6 with CE81T2-4 cell line, the further treatment of recombinant elafin can reverse the expression of knockdown-induced markers of mesenchymal-to-epithelial transition in a time-dependent manner (Figure 5E). The above reversed expression of EMT was also found in the CE81T2 cell line when recombinant elafin was added (Figure 5F). These results indicated that the exposure to exogenous elafin promoted tumor cell migration and invasion ability through its effect on the EMT pathway.

## 4. Discussion

Using 3-D tissue images, we found that there were two distinctly different distribution patterns of elafin protein in almost the same clinical stages of locoregional ESCC patients, and that each of these distributions led to different prognostic consequences. We also found elafin promoted ESCC tumor cell migration and invasion through the pathway of EMT gene expression. This study adds further insight into the biological mechanisms underlying the relationship between serum elafin and poor prognosis in locoregional ESCC patients [5] and it augments previous findings of elafin function using panoramic 3-D images.

Elafin, a secretory small molecule constitutively produced by skin squamous epithelium, forms the antimicrobial complex on mucosal and epithelial surfaces [9,10,11]. Studying the 3-D tissue structures of two ESCC patients who had higher serum levels of elafin and poor prognosis, we found elafin to be distributed in an interweaved-like fibrous structure in stroma, but not in the tumor region. Because elafin has previously been found to be able to crosslink with other extracellular matrix proteins, including lamin and elastin, via transglutaminisation on repeated VKGQ sequence in the N-terminal domain [20], we speculate that elafin’s interweaved fibrous structure may indicate elafin was released to stroma and resulted in the elevation of elafin levels in serum. By contrast, the 3-D imaging of the tissue structures of another two ESCC patients who had lower serum elafin and good prognosis showed elafin expression to be confined inside or around the tumor nest instead of the stroma. Elafin has also been reported to serve as a scaffold for crosslinking small proline-rich proteins during the formation of cornified envelopes in the squamous epithelium [21,22]. The confined elafin expressions, possibly due to the keratinization of well-differentiated ESCC inside the tumor nest, caused low elafin levels in serum [7,8]. In this study, we only performed panoramic tissue imaging of elafin expression in two pairs of ESCC patients. Further studies are needed better understand the molecular mechanisms underlying elafin’s escape from the confines of the tumor nest and diffuse infiltration into the stroma, adversely affecting the prognosis of locoregional ESCC.

3-D optical tissue images can characterize tumor microenvironments better than conventional 2-D IHC staining. In this study, the 3-D images provided the optical data for tissues 130–150 μm thick equaling a depth of ~13–15 cells, while 2-D IHC only provided the image of tissues 3–5 μm thick—the thickness of a half-cell only—a limitation that might lead to some misinterpretation of elafin signals in tumor tissue slides. Thus, the use of 2-D imaging in most studies might explain the reason that, to date, it is undecided whether elafin plays an oncogenic role or a tumor-suppressive role in different human cancers (Appendix A). A few studies of ESCC have found higher elafin protein expression in tumor nests with well-differentiated tumor cells, but not in highly proliferated PCNA-expressing tumor cells [7,23]. Their IHC results indicated that elafin expression in tumor is associated with a favorable prognosis, findings that conflict with our recent results showing that high serum elafin levels predict poor prognosis of locoregional ESCC [5]. The conflicting results are probably due to their use of 2-D IHC staining, which is unable to comprehensively evaluate elafin expression in 3-D tumor tissues [7,23]. Indeed, consistent with our previous findings, the 3-D panoramic images in the current study clearly show that confined elafin signals in the tumor nest are associated with a favorable prognosis, while interweaved fibrous elafin throughout the stroma is associated with a poor prognosis.

The mechanisms underlying elafin’s regulation of cellular physiology is complex and the role of elafin in cancer progression is still controversial. One previous study found that elafin inhibited neutrophil elastase-induced cell growth of human mammary epithelial cells, a process found to be mediated by ERK signaling [24]. By contrast, elafin has been found to promote cell proliferation through its activation of the MAP kinase pathway, and it has been associated with chemoresistance via Bcl-Xl expression in high-grade serous ovarian cancers and basal-like breast tumors [25]. In one study of SKOV3 ovarian cancer cells, knocking down elafin increased apoptosis after cisplatin treatment and the overexpression of elafin led to increased cisplatin resistance [26]. However, little is known about its biological mechanism in ESCC.

The current study found that the overexpression of elafin provoked the proliferation, migration, and invasion of the ESCC cell lines, and knocking down elafin by recombinant elafin reversed these three effects. Our pathway analysis of whole-gene expression in both overexpressed and knocked-down elafin revealed that elafin was involved in cell migration and invasion, and affected EMT-related gene expression. Indeed, our in vitro studies further showed that the expressions of EMT-related protein markers such as Snail, Slug, N-cadherin, and E-cadherin were also influenced in the presence of elafin. Previous clinical and preclinical studies have found EMT pathways to be related to cell invasion, distant metastatic formation, and poor outcomes or recurrence [27,28]. EMT, the first in a series of steps involved in metastasis, causes tumor cells to disassociate and prevent anoikis in the blood stream [29]. To the best of our knowledge, this is the first study to find that elafin regulates tumor cell EMT, possibly through its mediation of Snail and Slug protein expression.

This study has several limitations. One limitation is that this study only performed 3-D tissue imaging for four representative ESCC patients. More subjects are required to elucidate the role of elafin on ESCC tumor tissues. Another limitation is that the patients for the IHC-staining tissue arrays did not have serum samples to measure elafin levels and survival data, so we were unable to examine the correlation of elafin expression in both tissue and serum and the effect of elafin expression in tissues on survival. Another limitation is that the detailed molecular mechanism underlying secretory elafin upregulation of the EMT pathway markers was not fully elucidated. Further study is needed to determine whether elafin binds to cell membrane receptors and triggers downstream signal transduction or if elafin is captured via endocytosis to trigger signal cascades inside the cytoplasm.

In conclusion, using different omics approaches, we found that differences in patterns of elafin distribution within ESCC tissue are associated with serum elafin levels and prognoses in ESCC patients with no distant metastasis. Elafin was found to influence ESCC motility and invasion via its effect on the regulatory proteins of EMT. Thus, the inhibition of elafin may be a potential therapeutic target of locoregional ESCC to improve survival.

## 5. Conclusions

This study used three-dimensional confocal imaging to examine the spatial distribution of elafin in the tumor tissues of locoregional esophageal squamous cell carcinoma (ESCC), and found distinct localizations of elafin between favorable and poor prognosis patients. We found elafin expression to be significantly higher in tumor parts than in normal parts of the collected tissues, and higher elafin contributes to poor prognosis. In vitro mechanistic studies found that elafin could promote ESCC cell proliferation, migration, and invasion via the epithelial–mesenchymal transition pathway. Because elafin can promote the aggressiveness of locoregional ESCC, we concluded that the inhibition of elafin could potentially be used therapeutically to improve survival in patients with locoregional ESCC.

## Figures and Tables

**Figure 1 cancers-15-03825-f001:**
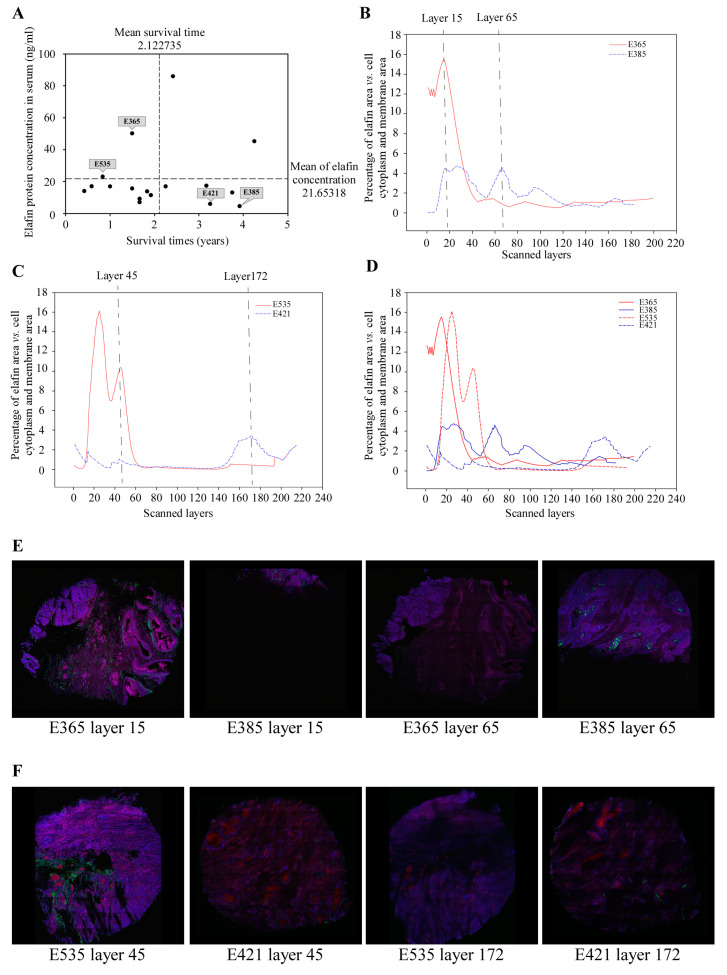
Elafin expressions in different layers of ESCC tumor tissues. (**A**) Patients sorted by mean serum elafin and mean survive time. (**B**) Spatial distribution of relative elafin expressions in tissue layers of Pair 1 (E365 and E385) from top to bottom. (**C**) Spatial distribution of relative elafin expressions in tissues layers of Pair 2 (E535 and E421) from top to bottom. (**D**) Combined data of Pair 1 and Pair 2. (**E**) Representative stitched 2-D confocal images of elafin expressions in two different tissue layers of Pair 1 (green for elafin, blue for cell membrane and cytoplasm, and red for cell nuclei). (**F**) Representative stitched 2-D confocal images of elafin expressions in two different tissue layers of Pair 2.

**Figure 2 cancers-15-03825-f002:**
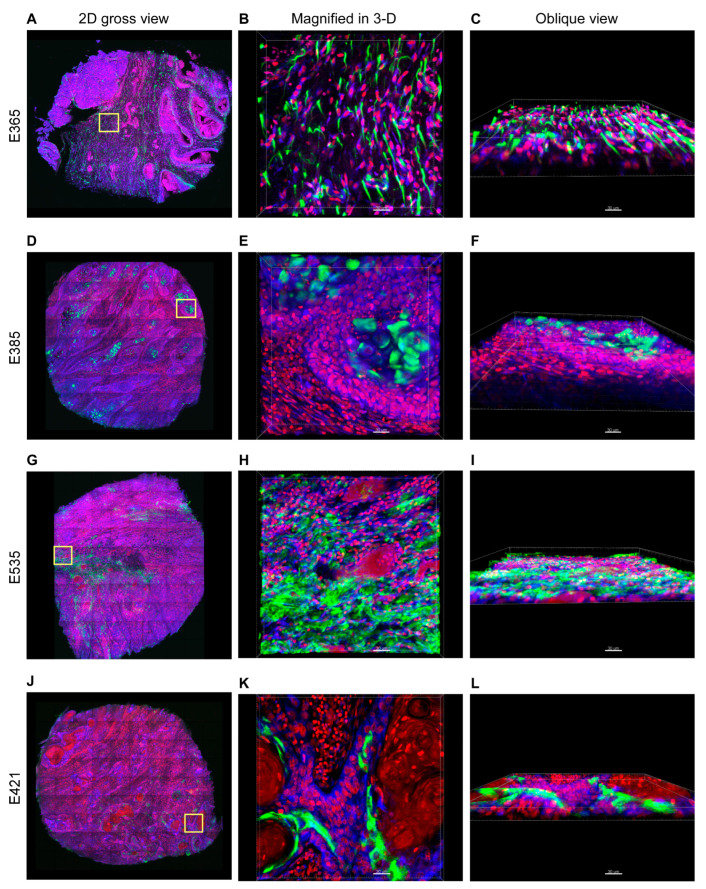
Projected gross 2-D tissue view and representative magnified 3-D tissue imaging of Pairs 1 and 2. (**A**) Gross 2-D tissue view of E365 projected from 199 tissue layers (139.3 μm in z axis) with the stitched tissue image (3261.49 μm in x axis; 2672.79 μm in y axis). (**B**) Magnified 3-D look-down view of yellow square from (**A**); the tissue volume of the yellow square was 317.95 µm × 317.95 µm × 45.5 μm. (**C**) The oblique view of (**B**). (**D**) Gross 2-D view of E385 projected from 182 tissue layers (127.4 μm in z axis) with the stitched tissue image (2672.78 μm in X axis; 2672.78 μm in y axis). (**E**) Magnified 3-D look-down view of yellow square from (**D**); the tissue volume of the yellow square was 317.95 µm × 317.95 µm × 76.3 μm. (**F**) The oblique view of (**E**). (**G**) Gross 2-D view of E535 projected from 190 tissue layers (133.0 μm in z axis) with the stitched image (2392.09 μm in X axis; 2980.17 μm in y axis). (**H**) Magnified 3-D look-down view of yellow square from (**G**); the tissue volume of the yellow square was 317.95 µm × 317.95 µm × 33.6 μm. (**I**) The oblique view of (**H**). (**J**) Gross 2-D view of E421 projected from 215 tissue layers (150.5 μm in z axis) with the stitched image areas (2968.38 μm in x axis; 2674.64 μm in y axis). (**K**) Magnified 3-D look-down view of yellow square from (**J**); the tissue volume of the yellow square was 317.95 µm × 317.95 µm × 14 μm. (**L**) The oblique view of (**K**). Green represents elafin, blue the cell membrane and cytoplasm, and red the cell nuclei; scale bar = 30 μm.

**Figure 3 cancers-15-03825-f003:**
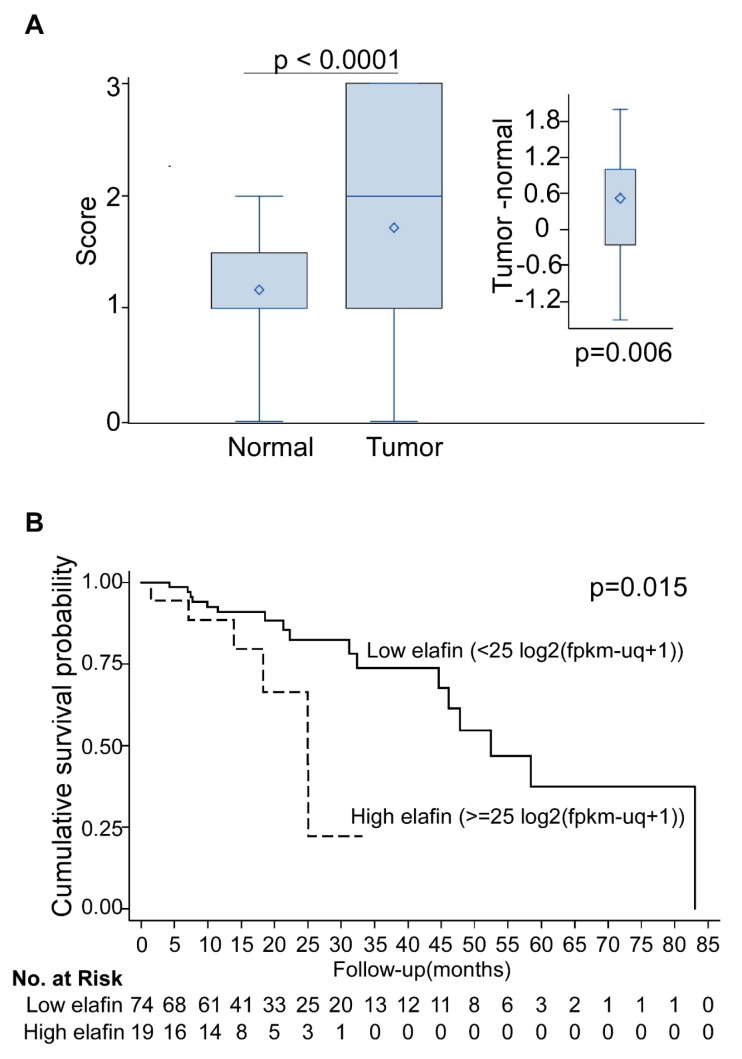
Elafin expression and clinical outcome of ESCC patients. (**A**) Elafin protein expression by immunohistochemistry staining in slide tissue arrays of 63 locoregional ESCC patients. Inset indicates 36 ESCC patient with both tumor and normal parts. (**B**) Kaplan–Meier survival curves of 93 locoregional esophageal cancer patients from TCGA database with elafin cutoff set at 25 (*p*-value = 0.015). Abbreviation: ESCC = esophageal squamous cell carcinoma.

**Figure 4 cancers-15-03825-f004:**
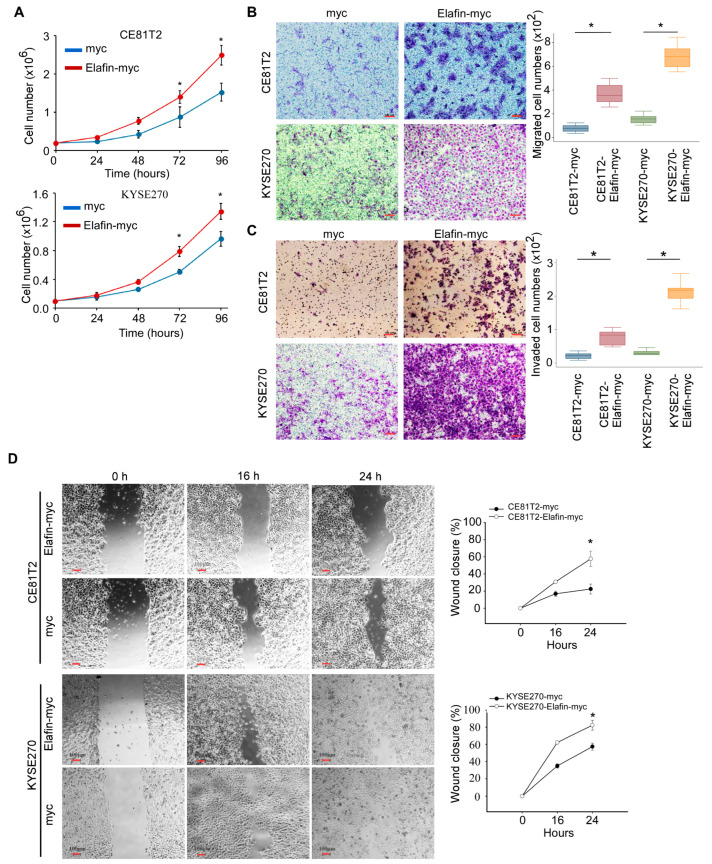
Ectopic elafin increased proliferation, migration, invasion, and wound-healing ability of ESCC cell lines (CE81T2 and KYSE270). (**A**) Ectopic elafin increased cell proliferation in a time-dependent curve. (**B**) Ectopic elafin increased cell migration. (**C**) Ectopic elafin increased cell invasion. (**D**) Ectopic elafin increased wound-healing ability. Scale bar = 100 μm. Abbreviation: ESCC = esophageal squamous cell carcinoma. Mean ± SD. * *p* < 0.05.

**Figure 5 cancers-15-03825-f005:**
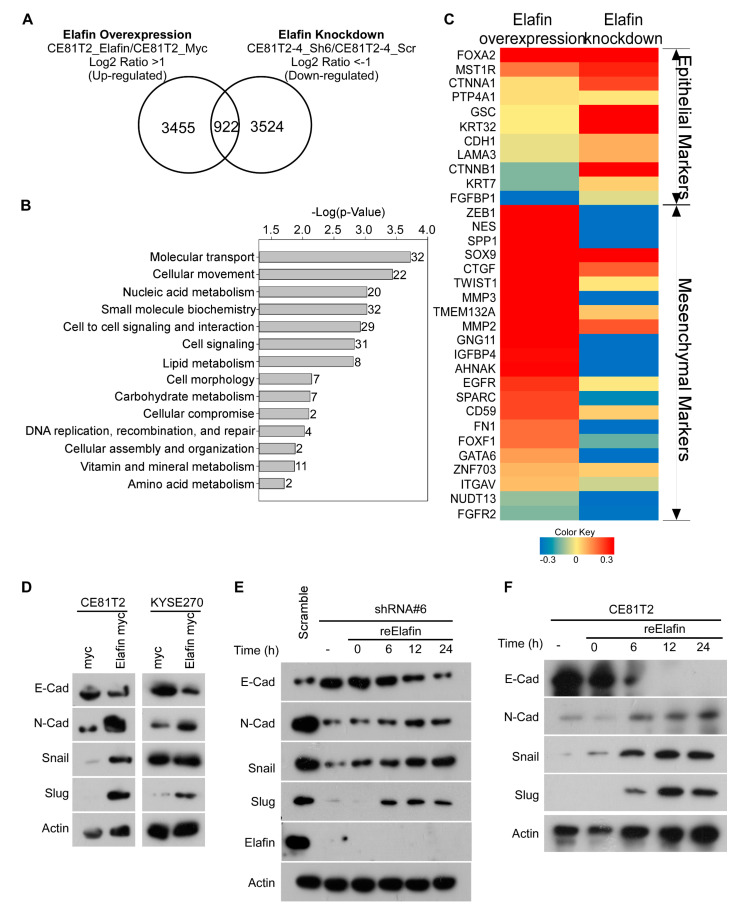
Elafin influenced cell migration and invasion by mediating the epithelial-to-mesenchymal transition (EMT)-related genes. (**A**) Number of overlapping genes involved in both elafin-overexpressed CE81T2 cell line and elafin-knockdown CE81T2-4 cell line by next-generation sequencing. (**B**) As evidenced by the pathway analysis of the IPA software, 209 out of 922 overlapping genes matched 14 pathways (the numbers indicate how many genes participated in a specific pathway, and the *p*-value represents the minimum value inside these genes). (**C**) Heatmap analysis of EMT-related genes. Both groups used the same nomenclature from (**A**). (**D**) Overexpression of elafin by elafin-myc altered EMT markers in CE81T2 and KYSE270 cell lines. (**E**) Restored EMT marker expression in elafin-specific shRNA knockdown CE81T2-4 cell line after treatment with rElafin over time. (**F**) Changes of EMT markers in CE81T2 cell line after treatment with rElafin over time. Abbreviations: IPA = ingenuity pathway analysis; rElafin = recombinant elafin. Original blots can be found in Appendix A.

**Table 1 cancers-15-03825-t001:** Baseline characteristics of two paired locoregional esophageal squamous cell carcinoma patients.

	Pair 1		Pair 2	
	E365	E385	E535	E421
Gender	Male	Male	Male	Male
Age	53	53	49	60
Stage	IIb	IIIa	IIIa	IIIa
Treatment	OP then CCRT	OP then CCRT	OP	OP
Serum elafin (ng/mL)	50.28	4.67	23.06	6.09
Survival time (Months)	18	47	10	39
Scanned layers ^a^ (Thickness, μm)	199 (139.3)	182 (127.4)	190 (133.0)	215 (150.5)
Layers of selection ^b^ (Thickness, μm)	1st–64th (45.5)	10th–118th (76.3)	13th–60th (33.6)	1st–20th (14.0)

Abbreviations: ESCC, esophageal squamous cell carcinoma; CCRT, concurrent chemoradiation therapy; OP, surgical operation of esophageal resection. ^a^ Different total tissue layers were scanned in different subjects due to different tumor volumes collected from the surgery. ^b^ Tissue layers with the relative elafin intensity > 0.01 were used to construct 3-D tissue structure.

## Data Availability

The datasets of next-generation sequencing of RNA-seq (Dataset 1) for this study can be found in the Gene Expression Omnibus database as GSE205623 (https://www.ncbi.nlm.nih.gov/geo/query/acc.cgi?acc=GSE205623). The rest of the data that support the findings of this study are available on request from the corresponding author.

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
