# Peer review of "Elafin as a Prognostic Marker in Esophageal Squamous Cell Carcinoma: A Pilot Study Using Three-Dimensional Imaging and Genomic Profiling"

_cancers, 2023, doi:10.3390/cancers15153825_

Round 1

Reviewer 1 Report

Elafin as a Prognostic Marker in Esophageal Squamous Cell Carcinoma: A Pilot Study using 3D Imaging and Genomic Profiling is a very  interesting paper aimed to explore the relationship of elafin with survival. Three-dimensional tissue imaging showed elafin distributed as an interweaved-like fibrous structure in the stroma of tissue obtained from patients with high serum levels of elafin and poorer prognoses. By contrast, the signal was confined inside or around the tumor nest in patients who had lower serum levels and better survival. The in-vitro studies revealed that elafin promoted ESCC cell proliferation, migration and invasion via the epithelial-mesenchymal transition pathway. The Authors concluded that, elafin inhibition could potentially be used therapeutically to improve survival in patients with locoregional Esophageal Squamous Cell Carcinoma.

I would just like to ask the Authors a question. The Authors stated that the cutoff value of the relative elafin intensity was set as 0.01 to filter out weak signals and select tissue layers with relative elafin intensity > 0.01 for further 3-D surface rendering.

What methodology was used to choose the cutoff value? Was a ROC curve performed?

Minor editing of English language required.

Reviewer 2 Report

In this study, the authors investigated the expression of Elafin and its correlation with clinical prognosis in oesophagal squamous cell carcinoma (ESCC). Using 3D fluorescent confocal microscopy and microscopic processing, the authors analysed the Elafin expression and distribution in a large volume of tumour specimens from two carefully selected matched pairs of ESCC patients. They found that increased expression with widespread distribution of Elafin was associated with poor prognosis. This finding was then verified in a large cohort of locoregional ESCC patients' microarray tissues and the clinical data from the TCGA database. Mechanistically, they showed by in vitro gain and loss of function studies that the enhanced Elafin was associated with activation of the EMT pathway, supported by cellular functions such as proliferation, migration and invasion, RNA-seq in conjunction with pathway analysis, and western blotting for the established EMT markers. Finally, the authors performed rescue studies with recombinant elafin, with the evidence backing up their findings. The study is comprehensive and interesting. The manuscript is well written, and the data are nicely presented in the figures and supplementary materials. Nevertheless, there are a few concerns about the current version of the manuscript.

Missing info: How long did the acquisition of each confocal image stack require roughly? This info is relevant to gauge potential photobleaching during this process. To address concerns about potential photobleaching during image acquisition, it would be ideal for the authors to demonstrate that photobleaching was not a significant issue in their study. It is also unclear how the coordination of series image acquisition was controlled, e.g. by semi-automated acquisition/machine-based learning or manual.

Figure 2/Pg9, line 318: the marker to identify the tumour nests should be used as in the 2D gross view, there is no indication where the tumour clusters are.

Pg4, lines 128-129: is it to use the Z projection tool to generate the compressed single 2D image? What is the ‘projection type’ used to generate the overall single intensity, the ‘sum of slices’? This detailed info is also relevant for the reader to understand image processing.   

Pg5, line 171: the gene name for elafin should be shown. 

Pg6, line 252: Fig S2 shows ~200 layers of the images acquired for each region. However, why are only 10 layers included here? Explain if there was no signal for the rest layers. Could it be due to image photobleaching?

Figure 1E: the 4th image is not correctly labelled

Remove the 1st paragraph in the Discussion

Typos/missing letters in places

Pg1, line 25: ‘his’ research

Pg4, line113 & 119 ‘X’ missing; line 266 ‘-‘; line 136: cell membrane member signal

Pg15, line 427: ‘stoma’

Fig S2 D: 307 µm x 307 µm?

The video display speed is a bit too fast.

Minor corrections are needed.

Reviewer 3 Report

Elafin is a secretory protein produced by epithelial esophageal cells, associated with the risk of ESCC. The tissue pattern of Elafin was related to the prognostic index of ESCC: a diffuse stromal pattern in ESCC with poor prognosis compared to peritumoral pattern in ESCC patients with better prognosis. In vitro studies have shown that Elafin promotes ESCC cell proliferation, migration, and invasion though the epithelial -mesenchymal transition pathway. Targeting of Elafin may have a potential role in innovative therapeutic approaches on ESCC.

The results are interesting and original and support q role for Elafin in both development and therapy in lung carcinoma.

Specific comments:

1) In the abstract the authors should mention the results obtained through the analysis of TCGA data on Elafin mRNA expression in ESCC.

2) It is unclear in the text whether ESCC samples investigated by IHC for Elafin protein tissue immunoreactivity can be subdivided in to a high and low score group and whether these groups would eventually show a different overall survival, as observed for TCGA patients subdivided into Elafin mRNA high and low groups.

3) Is there any difference in the genomic profile if Elafin-high and Elafin-low groups?
